# Characteristics of Pruritus in Various Clinical Variants of Psoriasis: Results of the Multinational, Multicenter, Cross-Sectional Study

**DOI:** 10.3390/life11070623

**Published:** 2021-06-27

**Authors:** Kamila Jaworecka, Dominika Kwiatkowska, Luiza Marek, Funda Tamer, Aleksandra Stefaniak, Magdalena Szczegielniak, Joanna Chojnacka-Purpurowicz, Monika Matławska, Ayla Gulekon, Jacek C. Szepietowski, Joanna Narbutt, Agnieszka Owczarczyk-Saczonek, Adam Reich

**Affiliations:** 1Department of Dermatology, Institute of Medical Sciences, Medical College of Rzeszow University, 35-055 Rzeszów, Poland; kamilajaworecka@gmail.com (K.J.); dominika.kwiatkowska.ur@gmail.com (D.K.); 2Department of Dermatology and Venerology, Faculty of Medicine, Ludwik Rydygier Collegium Medicum in Bydgoszcz, Nicolaus Copernicus University in Torun, 85-094 Bydgoszcz, Poland; Lui06@interia.pl; 3Department of Dermatology, Gazi University School of Medicine, 06560 Ankara, Turkey; fundatmr@yahoo.com (F.T.); gulekona@gazi.edu.tr (A.G.); 4Department of Dermatology, Venerology and Allergology, Wroclaw Medical University, 50-368 Wrocław, Poland; aleksandraannastefaniak@gmail.com (A.S.); jacek.szepietowski@umed.wroc.pl (J.C.S.); 5Department of Dermatology, Pediatric Dermatology and Oncology, Lodz Medical University, 91-347 Łódź, Poland; magda.szczegielniak@gmail.com (M.S.); joanna.narbutt@umed.lodz.pl (J.N.); 6Department and Clinic of Dermatology, Sexually Transmitted Diseases and Clinical Immunology, Faculty of Medicine, Collegium Medicum, University of Warmia and Mazury in Olsztyn, 10-959 Olsztyn, Poland; joannachojnacka2@wp.pl (J.C.-P.); mrs.matlawska@gmail.com (M.M.); aganek@wp.eu (A.O.-S.)

**Keywords:** psoriasis, palmoplantar pustulosis, pruritus, itch, itching

## Abstract

Psoriasis is a chronic, inflammatory skin disease present in about 3% of the world’s population. The clinical symptoms manifest diversely, therefore one can distinguish several subtypes of psoriasis. The majority of patients with psoriasis experience pruritus, which is an unpleasant sensation that decreases patients’ quality of life. The knowledge on pruritus in different subtypes of psoriasis is limited. We have performed a cross-sectional, prospective, and multicenter study to evaluate the relationship between clinical subtypes of psoriasis (large-plaque, nummular, guttate, palmoplantar, inverse, erythrodermic, palmoplantar pustular, generalized pustular psoriasis, and psoriasis of the scalp) and the prevalence, intensity, and clinical manifestation of itch. We introduced a questionnaire assessing various aspects of pruritus to a total of 254 patients. Out of these, 42 were excluded. Pruritus was present in 92.9% of the remaining patients and its prevalence did not depend on the clinical subtype. A correlation between the severity of psoriasis and the intensity of itch was explicitly noticeable in palmoplantar pustular psoriasis and scalp psoriasis (*p* < 0.05). The itch sensation was individual and differed among subtypes of psoriasis. In conclusion, pruritus is a frequent phenomenon, and its presentation is different in various subtypes of psoriasis.

## 1. Introduction

Psoriasis is a chronic, inflammatory skin disease, characterized by a multifactorial and complexed pathogenesis, and with a varied clinical picture. The most representative and common skin lesion is an erythematous plaque covered with silvery scales [1]. However, it is not the only one clinical manifestation of this disease. A less common subtype is pustular psoriasis, characterized by the formation of sterile pustules on an erythematous base [2]. This form may affect the entire body (i.e., generalized pustular psoriasis), or may be limited to the hands and feet (i.e., localized pustular psoriasis) [3]. Nowadays, it is widely recognized that pruritus is an important subjective symptom of psoriasis that affects up to 90% of patients [4]. This phenomenon is described as the most disturbing symptom of psoriasis, which has a negative impact on patients’ quality of life. The intensity of the itch seems to correlate with various psychological aspects including depressive symptoms or anxiety. However, data regarding factors that influence the intensity of pruritus and the presence of other symptoms associated with itching are still sparse. Moreover, the molecular basis of pruritus in psoriasis is still not fully elucidated, albeit a complex interaction between the nervous, neuroendocrine, immune, and vascular systems is suggested. Many mediators were indicated to modulate this sensation in psoriasis, but none has been proven to be a crucial one to date. The knowledge on pruritus in psoriasis is even more limited when considering different clinical subtypes of this disease, as most reports published so far have focused on plaque-type psoriasis. As a consequence, there is a challenge for clinicians to choose the most effective antipruritic treatment for specific subgroups of psoriatic patients [5]. The aim of this study was to evaluate the relationship between clinical subtypes of psoriasis and the prevalence, intensity, and clinical manifestation of itch.

## 2. Materials and Methods

### 2.1. Study Design

This was a cross-sectional, prospective study conducted in five Polish (Bydgoszcz, Łódź, Olsztyn, Rzeszów, Wrocław) and one Turkish (Ankara) dermatology clinics. Demographic data, anthropometric measurements, data on comorbidities including psoriatis artritis (PsA), and current treatment as well as clinical characteristics of pruritus were introduced into a questionnaire, which has been used by our group in previous studies [6]. Patients above 16 years old, with the confirmed diagnosis of psoriasis, were considered in the analysis. The exclusions criteria encompassed systemic antipsoriatic treatment within 4 weeks before the assessment, topical antipsoriatic treatment within 2 weeks before the assessment, other concomitant dermatological or systemic disorders that might cause pruritus such as chronic uremia or cholestasis, and usage of medications that could influence the sensation of itch. Disease severity was measured with Body Surface Area (BSA) and Static Physician Global Assessment (sPGA) in all patients, as well as with the Psoriasis Area and Severity Index (PASI) in patients with psoriasis vulgaris [7,8]. Patients with palmoplantar pustular psoriasis (PPPP) were assessed with Palmoplantar Pustulosis Severity Index (PPSI), while those suffering from generalized pustular psoriasis (GPP) were evaluated according to Generalized Pustular Psoriasis Severity Index (GPPSI) [9,10]. The maximal peak pruritus and the average pruritus intensity within the previous 3 days were evaluated with the 11-point Numerous Rating Scale (NRS_max_, and NRS_average_, respectively) from 0 (no pruritus) to 10 (worst imaginable pruritus) [11]. In addition, the 10-item Pruritus Severity Scale (10-PSS) was used as another method of measurement for pruritus intensity [12]. Regarding 10-PSS, the extension, intensity, and duration of pruritus episodes and their influence on concentration and psyche are evaluated, along with scratching as a result of pruritus. The maximal possible result is 20 points, which indicates the most severe itch. The Dermatology Life Quality Index (DLQI) was applied to evaluate the impact of skin lesions on the patient’s quality of life [13]. The DLQI scoring ranges from 0 to 30, with a higher score indicating a more impaired quality of life.

### 2.2. Patients

A total of 254 patients with different clinical subtypes of psoriasis were included. In most subjects, the diagnosis was made according to clinical manifestation, while in doubtful cases, a histopathological assessment of skin biopsies was performed. Forty-two individuals were excluded from further analysis because they were on systemic or topical treatment while data collection was performed. The remaining patients were divided into 9 groups according to the dominant subtype of psoriasis: large-plaque psoriasis (*n* = 45), nummular psoriasis (*n* = 27), guttate psoriasis (*n* = 24), palmoplantar psoriasis (*n* = 11), psoriasis of the scalp (*n* = 23), inverse psoriasis (*n* = 11), erythrodermic psoriasis (*n* = 18), PPPP (*n* = 42), and GPP (*n* = 11) (Table 1). The mean age of all patients was 45.1 ± 14.9 years (range: 16–77 years). Subjects with guttate psoriasis and scalp psoriasis were slightly younger, while those with pustular forms of psoriasis were slightly older than the rest of the patients, albeit the difference was significant (*p* < 0.001) (Table 1). Out of all studied subjects, 50% were women and 50% were men. However, in the erythrodermic subtype, men predominated (72.2%). On the contrary, the majority of patients with PPPP were women (85.7%; *p* < 0.001). The mean duration of psoriasis was 13.6 ± 12.3 years (range: 0–59 years) and this parameter was similar among all subtypes of psoriasis (Table 1). All patients agreed to take part in this study and signed the written informed consent. The study was approved by the Ethics Committee by Subcarpatian Physician Chamber in Rzeszów and also by the Gazi University Ethics Committee.

### 2.3. Statistical Analysis

All data were analyzed statistically with Statistica 13.0 (Statsoft, Krakow, Poland). Means, standard deviations (SD), median values, and frequencies were calculated. The differences between the groups of patients were analyzed using the Student’s *t*-test for independent variables, Mann–Whitney U test, and analysis of variance (ANOVA), where appropriate. Correlations between analyzed parameters were verified by using Spearman’s rank correlation test (ρ—correlation coefficient). χ^2^ test was used to determine whether there was a significant difference between the expected and observed frequencies in one or more categories. The results were considered statistically significant if the *p*-value was less than 0.05.

## 3. Results

### 3.1. Prevalence and Intensity of Pruritus

Pruritus ever in the course of psoriasis (the lifetime prevalence of pruritus) was experienced by the great majority (92.9%) of patients, while 81% reported the presence of this symptom within the 3 days preceding the examination (the point prevalence of pruritus). There were no significant differences between the clinical subtypes of psoriasis and the presence of itch in the course of the disease (*p* = 0.48) and within the last 3 days (*p* = 0.9). In most psoriasis subtypes, pruritus was limited to the lesional skin. Only in erythrodermic psoriasis was generalized pruritus observed with significantly higher frequency as compared to other subtypes (*p* = 0.03). Nonetheless, this result may not be so relevant due to the generalized distribution of skin lesions in the erythrodermic subtype of psoriasis. No significant relationships were found between subtypes of psoriasis and the intensity of pruritus according to NRS or 10-PSS. The detailed results referring to all groups are shown in Table 2.

### 3.2. Disease Severity

The mean PASI, BSA, sPGA, PPSI, and GPP scores in each subgroup are summarized in Appendix A. In scalp psoriasis and PPPP, the intensity of pruritus strongly correlated with the severity of psoriasis (*p* < 0.05). Interestingly, in nummular psoriasis, such a relationship was observed only for the 10-PSS score. A significant correlation was also seen between the severity of pruritus and BSA in GPP. In large plaque-type, guttate, palmoplantar, inverse, and erythrodermic psoriasis, no significant correlations were found between the intensity of pruritus and disease severity. Correlations between pruritus intensity and disease severity according to the morphological phenotype of psoriasis are presented in Table 3.

### 3.3. Descriptions of Pruritus

The perception of itch was quite individual and differed even within the same subtype of psoriasis. Most of these differences did not reach statistical significance. However, in both palmoplantar psoriasis and PPPP, patients more often described pruritus as painful and deep (*p* < 0.05). In addition, a burning or warming feeling accompanied 36% of patients with palmoplantar psoriasis. Furthermore, it seems that in GPP, itching can be experienced as feeling cold (*p* < 0.002). Nonetheless, there was a small population size and only one subject used this term to describe pruritus; thus, this result may be random. The characteristics of pruritus among different subtypes of psoriasis are presented in Appendix A.

Patients with scalp, erythrodermic, and inverse psoriasis experienced marked emotional irritation due to pruritus (*p* < 0.01). Patients also frequently mentioned that pruritus is quite disturbing (Table 2). We observed that in GPP, the most intense pruritus occurred when the skin lesion extended their size, and this feature was rather unique to this disease subtype. Frequency of pruritus and moment of alleviation of this symptom did not differ significantly between the groups. In each studied population, patients experienced itching every day or at least a few times a week, and pruritus was mostly relieved after the significant improvement or complete disappearance of skin lesions. Subjects with palmoplantar and inverse psoriasis had the most intense pruritus when skin lesions were fully developed (Appendix A).

### 3.4. Quality of Life

According to the collective analysis of all psoriatic patients, pruritus correlated positively with impaired quality of life. This correlation was strongly significant in plaque psoriasis and PPPP. In guttate or scalp psoriasis, the impact of itching on quality of life was also observed. Contrarily, such correlation was not observed in nummular, palmoplantar, inverse, erythrodermic psoriasis, and GPP (Table 3).

## 4. Discussion

Pruritus defined as an unpleasant sensation that provokes a desire to scratch was observed in 93% of all patients in our study. In other research, this phenomenon was reported with a frequency of 63–98% [14,15,16,17,18,19]. These differences may be due to regional features, the clinical characteristics of patients, and the divergent study methodologies used in other centers. To date, pruritus in psoriasis has already been investigated several times, but researchers have not distinguished any clinical subtypes of psoriasis or have mainly been focused on plaque-type psoriasis [20,21,22]. However, psoriasis manifests with several morphological phenotypes that may differ regarding the intensity and perception of pruritus. In the literature, descriptions of both prevalence and the clinical characteristics of pruritus in psoriasis are limited and contradictory [23,24,25]. Here, we have presented the results of a multicenter, prospective study analyzing pruritus in various morphological subtypes of psoriasis.

In our study, there was no relationship between the subtypes of psoriasis and the frequency of pruritus. Compatible results were observed by Sung-Min Park with his group, albeit assessment was performed only in guttate and plaque subtypes [16]. Similarly, Szepietowski et al., Yosipovitch et al., and Bahali et al. found that the presence of itch did not depend on the type of psoriasis, although the number of considered clinical subtypes were limited [18,26,27]. In contrast, another research team noticed some differences in the prevalence of pruritus. The most pruritogenic variant according to Sampogna et al. was palmoplantar psoriasis, whereas the least pruritogenic was guttate psoriasis (67.6% and 50.0%, respectively) [14]. On the other hand, two other studies suggested that the presence of pruritus was mostly linked to the pustular subtype [28,29].

Regarding localization of pruritus, we paid attention to whether pruritus was limited to lesional skin or also concerned uninvolved skin. In most psoriasis variants, pruritus was limited to psoriatic lesions. Contrarily, in one previously conducted study, itch was not limited to involved areas [18]. These divergent results may be caused by several limitations of Yosipovith’s trial: smaller studied group (*n* = 101), only three differentiated subtypes (plaque, guttate, erythrodermic), and 45% of subjects receiving antihistamines [18]. In our study, the only exception was erythrodermic psoriasis, in which generalized pruritus was more prevalent. However, it can be easily explained by specific skin involvement in erythrodermic psoriasis, which affects more than 80% of the skin [30].

The data on the interplay between psoriasis severity and itch remains inconclusive–some investigators have shown a positive connection between these factors, while others reported a lack of correlation [6,18,29,31]. However, even if the analysis was based on some different subtypes of psoriasis, the severity of the disease was assessed collectively [18,27]. In our groups of examined patients, we observed that the intensity of pruritus increased along with disease severity, but statistical significance was confirmed only in some disease variants. The most prominent relationship was found between the intensity of pruritus and the severity of the scalp psoriasis and PPPP. In nummular psoriasis, only the 10-PPS score correlated positively with psoriasis severity. However, such correlation was not significant according to the NRS_max_ and NRS_average_ scores. This result can be explained by the complex and multifactorial assessment of pruritus in the 10-PPS scale, which allowed for a more detailed evaluation of the intensity of pruritus along with the extension, duration, and the influence of pruritus on concentration, psyche, and scratching. Furthermore, in GPP, the extension of skin lesion measured by BSA was linked to higher pruritus intensity. The GPP index showed consistent results (ρ ranged from 0.31 to 0.63), albeit probably due to the small studied group, these results were not always statistically significant (*p* = 0.04–0.16).

It is difficult to uniformly characterize itch; even patients in the same subtypes of psoriasis used various terms to describe their feelings. The most common associated symptoms with large plaque psoriasis were burning and pure itch. Likewise, tingling, pinching, and painful pruritus were commonly used by the patients. Published data regarding individual patients’ sensations of itch seems to be limited. In one study, tickling, crawling, and burning were the terms most often used by patients to name pruritus. In this research, plaque, guttate, and erythrodermic subtypes of psoriasis were taken into consideration; unfortunately authors did not provide a separate analysis for certain types. It is worth mentioning that most of these patients suffered from plaque psoriasis (*n* = 92), while guttate and erythrodermic subtypes were in the great minority (*n* = 5 and *n* = 4, respectively); thus, one can assume that this result is related mostly to plaque psoriasis [18]. In our study, painful itch accompanied mainly palmoplantar types of psoriasis, and it was probably due to a higher density of free nerve endings in this area and the pressure causing additional pain [32]. On the other hand, 27% of responders with GPP also declared painful itch, which could suggest that different factors related to the formation of pustules in psoriasis may be relevant.

In most subtypes, itching occurred during the appearance of skin lesions (10–49%) or when they were fully developed (13–60%). The only exception was GPP, in which the highest intensity of pruritus was during the extension of the skin lesion. It is difficult to compare this data with previous studies as in many cases, no distinction of psoriasis subtypes was provided [6].

It has been well demonstrated that pruritus has a major impact on patients’ quality of life [33,34,35]. In concordance with previous studies, our data indicated that psoriatic patients suffering from intensive pruritus had significantly impaired quality of life [36,37]. Interestingly, in erythrodermic psoriasis, a lower quality of life did not depend on pruritus, which means that pruritus was not the major factor that impacted patients’ quality of life. This can be explained by the severe course of this disease subtype.

The pathogenesis of pruritus in psoriasis still remains unclear. Several potential mediators have been proposed, such as substance P, calcitonin gene-related peptide, neuropeptide Y, nerve growth factor and its receptors, lipocalin 2, and interleukin 31; however, data remain controversial and inconclusive [5,15,16,38,39,40]. Other authors suggested disturbances in the endogenous opioid system homeostasis, but there is still no final proof that indeed the observed abnormalities are a primary phenomenon in psoriasis leading to pruritus [41,42]. Recent data also indicated the role of transient receptor potential (TRP) channels, such as TRPV1, TRPV4, or TRPM8, but further studies are needed to confirm their relevance [43,44,45].

## 5. Limitations

This study has some limitations. Firstly, the number of patients in this study was not equal in each subtype of psoriasis. Additionally, there were only 11 patients in each of the groups for palmoplantar, inverse, and generalized pustular psoriasis, which makes statistical analysis difficult to interpret.

## 6. Conclusions

This study is unique because it is the first multicenter and prospective study characterizing pruritus in certain clinical variants of psoriasis. A better understanding of this phenomenon according to distinguished subtypes of psoriasis will help to improve communication between physicians and patients, adjust treatment to main complaints, and as a result, improve patients’ quality of life. A continuation of this research is needed to increase the sample size and confirm our findings.

## Figures and Tables

**Table 1 life-11-00623-t001:** Characteristics of included patients (* *p* values according to analysis of variance, ** *p* values according to χ^2^).

	**All** **Patients**	**Large-Plaque Psoriasis**	**Nummular** **Psoriasis**	Guttate Psoriasis	Palmo-Plantar Psoriasis	Psoriasis of the Scalp	Inverse Psoriasis	Erythrodermic Psoriasis	Palmo- Plantar Pustular Psoriasis	Generalized Pustular Psoriasis
Number of patients (%)	212 (100)	45 (21.2)	27 (12.7)	24 (11.3)	11 (5.2)	23 (10.8)	11 (5.2)	18 (8.5)	42 (19.8)	11 (5.2)
Age (years, mean ± standard deviation)	45.1 ± 14.9	46.8 ± 16.0	41.2 ± 12.4	37.4 ± 9.6	42.2 ± 12.6	34.6 ± 12.9	42.1 ± 15.3	47.7 ± 16.1	53.8 ± 12.8	54.1 ± 14.4
		*p* < 0.001 *								
Women (%)	106 (50)	15 (33.3)	10 (37)	10 (41.7)	5 (45.5)	13 (56.5)	5 (45.5)	5 (27.8)	36 (85.7)	6 (54.5)
Men (%)	106 (50)	30 (66.7)	17 (63)	14 (58.3)	6 (54.5)	10 (43.5)	6 (54.5)	13 (72.2)	6 (14.3)	5 (45.5)
		*p* < 0.001 **								
Duration of psoriasis (years)	13.6 ± 12.3	16.4 ± 13.4	14.8 ± 11.5	12.9 ± 10.2	10.2 ± 8.4	9.8 ± 8.4	11.7 ± 10.2	16.3 ± 14.2	8.3 ± 11.9	14.2 ± 11.5
		*p* = 0.06 *								

**Table 2 life-11-00623-t002:** Prevalence, intensity, and feelings related to pruritus (NRS—Numerical Rating Scale, 10-PSS—10-item Pruritus Severity Scale, SD—standard deviations, * *p* values according to χ^2^ test, ** *p* values according to analysis of variance).

	**All** **Patients**	**Large-Plaque Psoriasis**	**Nummular** **Psoriasis**	Guttate Psoriasis	Palmoplantar Psoriasis	Psoriasis of the Scalp	Inverse Psoriasis	Erythrodermic Psoriasis	Palmo-Plantar Pustular Psoriasis	Generalized Pustular Psoriasis	*p*
Pruritus Prevalence											
Pruritus present ever during the disease course: *n* (%)	197 (92.9)	43 (95.6)	26 (96.3)	22 (91.7)	10 (90.9)	23 (100)	10 (90.9)	16 (88.9)	36 (85.7)	11 (100)	0.48 *
Pruritus present within the last 3 days: *n* (%)	172 (81.1)	36 (80)	23 (85.2)	19 (79.2)	9 (81.8)	21 (91.3)	9 (81.8)	15 (83.3)	31 (73.8)	9 (81.8)	0.9 *
Pruritus limited to skin lesions: *n* (%)	159 (80.7)	35 (81.4)	23 (88.5)	21 (95.5)	8 (80)	18 (78.3)	7 (70)	7 (43.7)	32 (80.9)	8 (72.7)	0.03 *
Pruritus also involving non-diseased skin: *n* (%)	11 (5.6)	1 (2.3)	1 (3.8)	0 (0)	2 (20)	2 (8.7)	0 (0)	2 (12.5)	2 (5.6)	1 (9.1)	
Generalized pruritus: *n* (%)	27 (13.7)	7 (16.3)	2 (7.7)	1 (4.5)	0 (0)	3 (13)	3 (30)	7 (43.8)	2 (5.6)	2 (18.2)	
Pruritus Intensity											
NRS_average_ (mean ± SD)	3.7 ± 2.7	4.2 ± 2.9	2.9 ± 2.0	3.7 ± 2.7	3.5 ± 2.9	3.6 ± 2.3	3.9 ± 3.1	4.1 ± 2.9	3.8 ± 2.6	3.5 ± 3.1	0.83 **
NRS_max_ (mean ±S D)	4.8 ± 2.9	5.1 ± 3.2	4.2 ± 2.6	4.7 ± 2.9	4.7 ± 3.4	4.8 ± 2.6	5.1 ± 3.3	4.9 ± 2.9	4.8 ± 2.9	4.7 ± 3.4	0.99 **
10-PSS (mean ± SD)	8.5 ± 4.5	9.7±4.4	7.6 ± 3.3	7.8 ± 4.5	8.4 ± 4.6	8.1 ± 4.1	8.0 ± 4.8	9.0 ± 5.9	8.5 ± 4.7	8.6 ± 5.7	0.79 **
Feelings Related to Pruritus											
Disturbing: *n* (%)	120 (60.9)	28 (65.1)	15 (57.7)	13 (59.1)	7 (70)	9 (39.1)	5 (50)	9 (56.2)	30 (83.3)	5 (45.5)	0.29
Irritating: *n* (%)	87 (44.2)	11 (25.6)	15 (57.7)	9 (40.9)	6 (60)	14 (60.9)	8 (80)	10 (62.5)	11 (30.1)	3 (27.3)	<0.01
Annoying: *n* (%)	40 (20.3)	12 (27.9)	3 (11.5)	3 (13.6)	2 (20)	3 (13)	3 (30)	5 (31.2)	5 (13.9)	4 (36.4)	0.33
Distressing: *n* (%)	27 (13.7)	4 (9.3)	5 (19.2)	1 (4.5)	0 (0)	3 (13)	1 (10)	4 (25)	7 (19.4)	2 (18.2)	0.54

**Table 3 life-11-00623-t003:** Correlations between pruritus intensity and disease severity according to the Spearman rank correlation test (statistically significant values marked in bold; 10-PSS—10-item Pruritus Severity Scale, BSA—Body Surface Area, DLQI—Dermatology Life Quality Index, GPPSI—Generalized Pustular Psoriasis Severity Index, NRS—Numerous Rating Scale, PASI—Psoriasis Area and Severity Index, PPSI—Palmoplantar Pustulosis Severity Index, sPGA—Static Physician Global Assessment).

		**All** **Patients**	**Large-Plaque** **Psoriasis**	**Nummular Psoriasis**	Guttate Psoriasis	Palmo-Plantar Psoriasis	Psoriasis of the Scalp	Inverse Psoriasis	Erythrodermic Psoriasis	Palmoplantar Pustular Psoriasis	Generalized Pustular Psoriasis
PASI vs.	NRS_average_	**ρ = 0.22, *p* < 0.01**	ρ = 0.26, *p* = 0.09	ρ = 0.16, *p* = 0.43	ρ=0.31, p = 0.14	ρ = 0.15, *p* = 0.69	**ρ = 0.65, *p* < 0.001**	ρ = 0.25, *p* = 0.47	**ρ = 0.51, *p* = 0.03**	-	-
	NRS_max_	**ρ = 0.17, *p* < 0.03**	ρ = 0.28, *p* = 0.07	ρ = 0.18, *p* = 0.38	ρ=0.15, p = 0.49	ρ = 0.13, *p* = 0.73	**ρ = 0.64, *p* < 0.001**	ρ = 0.25, *p* = 0.45	**ρ = 0.54, *p* = 0.02**	-	-
	10-PSS	**ρ = 0.23** ***p*** **< 0.01**	ρ = 0.19, *p* = 0.24	**ρ = 0.52, *p* < 0.01**	ρ=0.09, p = 0.7	ρ = 0.46, *p* = 0.25	**ρ = 0.74, *p* < 0.001**	ρ = 0.02, *p* = 0.95	ρ = 0.41, *p* = 0.1	-	-
BSA vs.	NRS_average_	**ρ = 0.18, *p* = 0.01**	ρ = 0.15, *p* = 0.33	ρ = 0.28, *p* = 0.16	ρ = 0.31, *p* = 0.14	ρ = −0.01, *p* = 0.99	**ρ = 0.68, *p* < 0.001**	ρ = 0.21, *p* = 0.54	ρ = −0.04, *p* = 0.88	**ρ = 0.35, *p* < 0.05**	**ρ = 0.85, *p* < 0.01**
	NRS_max_	**ρ = 0.16, *p* = 0.02**	ρ = 0.17, *p* = 0.28	ρ=0.26, p = 0.2	ρ=0.23, p = 0.28	ρ = −0.1, *p* = 0.78	**ρ = 0.71, *p* < 0.001**	ρ = 0.22, *p* = 0.52	ρ = −0.13, *p* = 0.61	**ρ = 0.38, *p* = 0.03**	**ρ = 0.88, *p* < 0.01**
	10-PSS	**ρ = 0.17, *p* = 0.02**	ρ = 0.08, *p* = 0.60	**ρ = 0.52, *p* < 0.01**	ρ=0.13, p = 0.57	ρ = 0.16, *p* = 0.69	**ρ = 0.63, *p* = 0.002**	ρ = 0.22, *p* = 0.6	ρ = −0.14, p = 0.6	ρ = 0.21, *p* = 0.25	**ρ = 0.75, *p* = 0.03**
sPGA vs.	RS_average_	**ρ = 0.35, *p* < 0.001**	ρ = 0.27, *p* = 0.08	ρ = 0.17, *p* = 0.4	ρ=0.22, p = 0.3	ρ = 0.35, *p* = 0.3	**ρ = 0.61, *p* = 0.002**	ρ = 0.42, *p* = 0.2	ρ = 0.32, *p* = 0.19	**ρ = 0.48, *p* = 0.001**	ρ = 0.4, *p* = 0.22
	NRS_max_	**ρ = 0.37, *p* < 0.001**	**ρ = 0.31, *p* = 0.04**	ρ = 0.28, *p* = 0.17	ρ=0.24, p=0.25	ρ = 0.48, *p* = 0.14	**ρ = 0.53, *p* < 0.01**	ρ = 0.4, *p* = 0.22	ρ = 0.39, *p* = 0.1	**ρ = 0.56, *p* < 0.001**	ρ = 0.59, *p* = 0.06
	10-PSS	**ρ = 0.4,** ***p* < 0.001**	ρ = 0.21, *p* = 0.19	**ρ = 0.63, *p* < 0.001**	ρ=0.11, p = 0.62	ρ = 0.49, *p* = 0.18	**ρ = 0.52, *p* = 0.01**	ρ = 0.3, *p* = 0.47	ρ = 0.36, *p* = 0.15	**ρ = 0.66, *p* < 0.001**	ρ = 0.34, *p* = 0.31
PPSI vs.	NRS_average_	-	-	-	-	-	-	-	-	**ρ = 0.46, *p* = 0.002**	-
	NRS_max_	-	-	-	-	-	-	-	-	**ρ = 0.53, *p* < 0.001**	-
	10-PSS	-	-	-	-	-	-	-	-	**ρ = 0.6, *p* < 0.001**	-
GPPSI vs.	NRS_average_	-	-	-	-	-	-	-	-	-	ρ = 0.31, *p* = 0.11
	NRS_max_	-	-	-	-	-	-	-	-	-	**ρ = 0.63, *p* = 0.04**
	10-PSS	-	-	-	-	-	-	-	-	-	ρ = 0.46, *p* = 0.16
DLQI vs.	NRS_average_	**ρ = 0.32, *p* < 0.001**	**ρ = 0.51, *p* < 0.001**	ρ = 0.005, *p* = 0.98	ρ = 0.13, *p* = 0.55	ρ = 0.16, *p* = 0.64	**ρ = 0.44, *p* = 0.03**	ρ = 0.12, *p* = 0.72	ρ = 0.22, *p* = 0.37	**ρ = 0.48, *p* = 0.001**	ρ = 0.58, *p* = 0.06
	NRS_max_	**ρ=0.35, *p* < 0.001**	**ρ = 0.56, *p* < 0.001**	ρ = −0.09, *p* = 0.66	ρ = 0.3, *p* = 0.15	ρ = 0.19, *p* = 0.58	ρ = 0.41, *p* = 0.05	ρ = 0.21, *p* = 0.54	ρ = 0.25, *p* = 0.31	**ρ = 0.57, *p* < 0.001**	ρ = 0.38, *p* = 0.25
	10-PSS	**ρ = 0.47, *p* < 0.001**	**ρ = 0.46, *p* = 0.003**	ρ = 0.17, *p* = 0.41	**ρ = 0.6, *p* = 0.002**	ρ = 0.39, *p* = 0.3	**ρ = 0.76, *p* < 0.001**	ρ = 0.47, *p* = 0.24	ρ = 0.34, *p* = 0.18	**ρ = 0.58, *p* < 0.001**	ρ = 0.44, *p* = 0.18

## Data Availability

Data is contained within the article or Appendix A.

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
