# Peer review of "Characteristics of Pruritus in Various Clinical Variants of Psoriasis: Results of the Multinational, Multicenter, Cross-Sectional Study"

_life, 2021, doi:10.3390/life11070623_

Round 1

Reviewer 1 Report

Go on collecting patient with less frequent subtype psoriasis 

Author Response

We are grateful to the reviewer for that comment. Indeed, we are continuing our study and we hope, that in the future we will be able to provide more detailed data. However, at the moment we would publish the study with already collected data. 

Reviewer 2 Report

An interesting questionnaire about pruritus administered to patients suffering from various subtypes of psoriasis, indicating a strong correlation between the symptom and the condition, as already reported in the medical literature. The statistically significant correlation between the severity of palmoplantar and scalp psoriasis and the intensity of itch is interesting, although the groups have a limited number of patients.

I have some queries:

page 3 line 108 statistical analysis is lacking.......please check and add the types of statistical tests used, the statistical program, its maker, and location.

page 1 lines 43-45 "Psoriasis is a chronic, inflammatory skin disease, characterized by amultifactorial and complexed pathogenesis, and with varied clinical picture. The most representative and common skin lesion is an erythematous plaque covered with silvery scales. " this sentence needs a reference, such as: doi: 10.3390/healthcare9050543.

Page 2 line 48 you should add: "this form may interest all the body or may be limited to the hands and feets. " and add a citation such as: doi: 10.1111/dth.13170. 

Table 4 should be better edited to fill in the margins of the sheet.

Was the study also approved by the local ethical committee in turkey?

Thank You

Author Response

We are grateful to the reviewer for his valuable comments. We have changed the original manuscript as follows:

Query: page 3 line 108 statistical analysis is lacking.......please check and add the types of statistical tests used, the statistical program, its maker, and location.

Authors: The subchapter on statistical analysis has been added. We apologize for this mistake. 

Query: page 1 lines 43-45 "Psoriasis is a chronic, inflammatory skin disease, characterized by multifactorial and complexed pathogenesis, and with the varied clinical picture. The most representative and common skin lesion is an erythematous plaque covered with silvery scales. " this sentence needs a reference, such as doi: 10.3390/healthcare9050543.

Authors: The respective reference has been added. 

Query: Page 2 line 48 you should add: "this form may interest all the body or may be limited to the hands and feets. " and add a citation such as: doi: 10.1111/dth.13170. 

Authors: The respective reference has been added. 

Query: Table 4 should be better edited to fill in the margins of the sheet.

Authors: Tables have been redesigned. 

Query: Was the study also approved by the local ethical committee in Turkey? Yes, it was. See attached document.

Reviewer 3 Report

The manuscript is interesting and well organized.
The authors could enrich the discussions and conclusions by making references to the possible molecular/genetic mechanisms involved in the different phenotypes.
The tables need to be reformulated as they are confusing and unclear.

Author Response

We are grateful to the reviewer for his very supportive words. We did the following modifications to the original manuscript: 

Query: The authors could enrich the discussions and conclusions by making references to the possible molecular/genetic mechanisms involved in the different phenotypes.

Authors: As suggested, we modified the discussion.

Query: The tables need to be reformulated as they are confusing and unclear.

Authors: Tables were redesigned.  

Reviewer 4 Report

In this article, Jaworecka et al. report their results on the multicenter and prospective study quantifying various characteristics of pruritus in distinct clinical variants of psoriasis. The authors present a lot of various data obtained from 254/212 patients and come to a general conclusion that pruritus is a frequent phenomenon and its presentation is different in various subtypes of psoriasis.

Overall the study is descriptive, unbiased and not hypothesis-driven. The data could be interesting for other clinicians or researches.

I have only several comments that should be addressed by the authors.

1) The text pertinent to the chapter 2.3 Statistical analysis is completely missing.

2) The authors may consider presenting some of their tables as a Supplementary Material. Instead, the main results should be summarized and qualitatively explained.

3) Table 1, which test was used fr this table?  

4) Table 2, I suppose that the last number at each row of the table is probability. Please format the table. Which test was used? What does the data mean?

5) Table 3, why are all these data in bold? The data comprising the Table 3 are not sufficiently described and explained. For example, second line (BSA), first column 22.2+/-26.1. What distribution characterizes these data (normal, if not, what range, median ?).  What should the reader infer from this Table?

6) Table 4, significant correlations are in bold. This should be written in the legend.

7) All references should be checked and formatted according to Journal guidelines.

Author Response

We are grateful to the reviewer for his valuable comments. We revised the manuscript to address all of them. following modifications were done.

Query: The text pertinent to the chapter 2.3 Statistical analysis is completely missing.

Authors: We apologize for this mistake. We have provided the description of statistical analysis. 

Query: The authors may consider presenting some of their tables as a Supplementary Material. Instead, the main results should be summarized and qualitatively explained.

Authors: We reconsidered and redesigned the tables. Some of them were included as supplemental material only. 

Query: Table 1, which test was used fr this table?  

Authors: We provided detailed information, which test was used for comparison.

Query: Table 2, I suppose that the last number at each row of the table is probability. Please format the table. Which test was used? What does the data mean?

Authors: We provided a more detailed description of this table. 

Query: Table 3, why are all these data in bold? The data comprising Table 3 are not sufficiently described and explained. For example, second line (BSA), first column 22.2+/-26.1. What distribution characterizes these data (normal, if not, what range, median ?).  What should the reader infer from this Table?

Authors: We have redefined and redesigned the tables to make the data more readable. 

Query:  Table 4, significant correlations are in bold. This should be written in the legend.

Authors: We provided respective explanations.

Query: 7) All references should be checked and formatted according to Journal guidelines

Authors: We apologize for the mistake, but the references were reformated by the program used to order the reference list. We have double-checked and corrected all references. 

Round 2

Reviewer 2 Report

The authors responded to all queries. The paper is publishable 

Reviewer 3 Report

The manuscript is carefully revised in accordance with the comments received.